# Anti-Angiogenic Therapy in *ALK* Rearranged Non-Small Cell Lung Cancer (NSCLC)

**DOI:** 10.3390/ijms23168863

**Published:** 2022-08-09

**Authors:** Aaron C. Tan, Nick Pavlakis

**Affiliations:** 1Division of Medical Oncology, National Cancer Centre Singapore, Singapore 169610, Singapore; 2Duke-NUS Medical School, National University of Singapore, Singapore 119077, Singapore; 3Department of Medical Oncology, Royal North Shore Hospital, Sydney, NSW 2065, Australia; 4Sydney Medical School, University of Sydney, Sydney, NSW 2006, Australia

**Keywords:** anaplastic lymphoma kinase, ALK rearrangement, anti-angiogenesis, non-small cell lung cancer, vascular endothelial growth factor

## Abstract

The management of advanced lung cancer has been transformed with the identification of targetable oncogenic driver alterations. This includes anaplastic lymphoma kinase (*ALK*) gene rearrangements. ALK tyrosine kinase inhibitors (TKI) are established first-line treatment options in advanced *ALK* rearranged non-small cell lung cancer (NSCLC), with several next-generation ALK TKIs (alectinib, brigatinib, ensartinib and lorlatinib) demonstrating survival benefit compared with the first-generation ALK TKI crizotinib. Still, despite high objective response rates and durable progression-free survival, drug resistance inevitably ensues, and treatment options beyond ALK TKI are predominantly limited to cytotoxic chemotherapy. Anti-angiogenic therapy targeting the vascular endothelial growth factor (VEGF) signaling pathway has shown efficacy in combination with platinum-doublet chemotherapy in advanced NSCLC without a driver alteration, and with EGFR TKI in advanced *EGFR* mutated NSCLC. The role for anti-angiogenic therapy in *ALK* rearranged NSCLC, however, remains to be elucidated. This review will discuss the pre-clinical rationale, clinical trial evidence to date, and future directions to evaluate anti-angiogenic therapy in *ALK* rearranged NSCLC.

## 1. Introduction

The management of advanced lung cancer has been transformed with the identification of targetable oncogenic driver alterations [1]. Notably, this includes anaplastic lymphoma kinase (*ALK*) gene rearrangements, which are detected in approximately 3–7% of patients with advanced non-small cell lung cancer (NSCLC) [2]. The most common gene fusion partner is echinoderm microtubule-associated protein-like 4 (*EML4*), which accounts for approximately 95% of *ALK* fusion variants, although a large range of other gene fusion partners have been reported [3]. *ALK* gene fusions lead to the aberrant oncogenic activation of the ALK tyrosine kinase [4]. Most *ALK* fusion variants in NSCLC have demonstrated sensitivity to ALK tyrosine kinase inhibitors (TKI) [3]. ALK TKI are established first-line treatment options in advanced *ALK* rearranged NSCLC, with several next-generation ALK TKIs (alectinib [5,6], brigatinib [7], ensartinib [8] and lorlatinib [9]) demonstrating survival benefit compared with the first-generation ALK TKI crizotinib in phase III randomized controlled trials. Consequently, there are an expanding number of oral targeted therapy options in the first-line treatment-naïve setting. Still, despite high objective response rates and durable progression-free survival, drug resistance inevitably ensues. Lorlatinib, a third-generation ALK TKI, has shown efficacy in the setting of resistance to prior first- and/or second-generation ALK TKI [10]. Beyond ALK TKI, however, treatment options remain predominantly limited to cytotoxic chemotherapy. Anti-angiogenic therapy, targeting the vascular endothelial growth factor (VEGF) signaling pathway, has shown efficacy in combination with platinum-doublet chemotherapy in advanced NSCLC without a driver alteration, and with EGFR TKI in advanced *EGFR* mutated NSCLC [11]. The role for anti-angiogenic therapy in *ALK* rearranged NSCLC, however, remains to be elucidated. This review will discuss the pre-clinical rationale, clinical trial evidence to date, and future directions to evaluate anti-angiogenic therapy in *ALK* rearranged NSCLC.

## 2. VEGF and Angiogenesis

The VEGF family consists of polypeptide growth factors including VEGF-A, -B, -C, -D, and -E [12]. VEGF-A is a key regulator of blood vessel development in adult tissues. VEGFs bind to receptors, consisting of three variants, VEGFR-1, -2 and -3 (Figure 1). This results in the formation of receptor homo- or heterodimers, activating downstream signaling pathways. VEGF expression is mediated by numerous growth factors, cytokines and hypoxia. In tumors, VEGF expressed by cancer cells can sustain tumor growth, commonly via the hypoxia-response pathway [13]. This pathway may be driven by hypoxia-inducible factors (HIFs) such as HIF-1, a heterodimer that binds to hypoxia-response elements and activates VEGF transcription. Given its key role in promoting angiogenesis and tumor proliferation, targeting VEGF with anti-angiogenic therapies has become a key strategy for cancer treatment [14].

## 3. Pre-Clinical Evidence for Anti-Angiogenic Therapy in NSCLC

### 3.1. Targeting the VEGF Axis in NSCLC

Targeting of the VEGF axis has a long history in advanced NSCLC. Bevacizumab, an anti-VEGF monoclonal antibody, was one of the first approved targeted therapies for non-squamous NSCLC nearly two decades ago, in combination with carboplatin plus paclitaxel chemotherapy [15]. Widespread adoption into the treatment paradigm for NSCLC was however limited by the modest survival benefits in conjunction with cost and drug toxicity considerations [16]. As the therapeutic landscape has evolved, with a growing number of targeted and immunotherapies leading to improved survival for patients with advanced NSCLC [17], the role for anti-angiogenic therapy is being revisited. This is further supported by a building body of pre-clinical evidence, suggesting that targeting of the VEGF axis may not only inhibit angiogenesis, but may also trigger anti-tumor immunity [18].

VEGF targeting therapies may exert broad anti-tumor and anti-angiogenic effects through direct inhibition of vessel growth inducing regional cancer cell death, induction of endothelial cell apoptosis through activation of BCL2 or Akt signaling, and stopping the recruitment of hematopoietic or endothelial progenitor cells for new vessel formation [19,20,21,22]. With an increased understanding of the importance of the tumor immune microenvironment, more recent studies have shown that VEGF inhibition may also have an immunomodulatory effect. A more immune permissive rather than immunosuppressive microenvironment may result from increased T-cell tumor infiltration and enhanced promotion of dendritic cell maturation [23,24]. Tumor endothelial cells (EC) may play a key role in this process. Anti-angiogenic therapy may restore proinflammatory surface proteins on tumor ECs through a process termed ‘vessel normalization’ [25]. This may not only improve immune cell activation and infiltration but may also enhance drug delivery into tumors with the formation of high endothelial venules (HEVs) [26,27]. Taken together, this has led to the evaluation of combinatorial approaches, particularly with immunotherapy (such as anti-PD-1/L1 inhibitors), and targeted therapies.

### 3.2. Pre-Clinical Evidence for ALK Rearranged NSCLC

For *ALK* rearranged NSCLC, there is a small body of pre-clinical evidence which supports the role for anti-angiogenic therapy. In a study by Watanabe et al. [28], combination ALK TKI (alectinib and crizotinib) with anti-VEGF2 therapy was evaluated in mouse xenograft models. Key findings included enhanced anti-tumor proliferative effects from the combination therapies compared with ALK TKI alone, induction of VEGFR2 expression after exposure to ALK TKI, implicating the role of VEGFR2 signaling with ALK TKI treatment, and lower numbers of CD31-positive blood vessels in mice treated with the combination treatment, suggesting anti-angiogenic efficacy. Importantly, VEGFR2 RNA expression was elevated in ALK-driven NSCLC cells, further increased after ALK TKI treatment, and decreased transiently after treatment. This implicates an important role for the VEGFR2 signaling pathway in *ALK* rearranged NSCLC. Further studies have demonstrated RAS-MAPK dependence as a hallmark of *EML4*-*ALK* rearranged NSCLC, with reactivation of the MAPK pathway associated with resistance to ALK inhibition [29]. Similarly, VEGF signaling may activate the MAPK pathway, as well as the PI3K/Akt pathway [30,31]. Another potential mechanism by which ALK and VEGF signaling may interact is through hypoxia pathways. Martinengo et al. [32] previously demonstrated that hypoxia pathways were significantly enriched in *ALK* rearranged NSCLC, compared with *EGFR* and *KRAS*-mutated NSCLC. ALK regulated VEGFA production and tumor angiogenesis in NSCLC, dependent on both HIF1α and HIF2α. Koh et al. [33] also demonstrated that ALK-translocated tumor cells may upregulate PD-L1 expression via enhanced HIF1α expression. Taken together, this suggests that dual blockade of ALK and VEGF may represent a rational combination approach. The potential cross-talk between the VEGF and ALK signaling pathways is shown in Figure 2. Combination approaches, however, should be cognizant of bioavailability and potential pharmacokinetic interactions. In particular, the drug efflux functions of P-glycoprotein and ATP-binding cassette super-family G member 2 (ABCG2) may be important for the bioavailability and distribution of ALK and VEGF TKIs [34,35,36,37].

Evidence is building on the role for anti-angiogenic therapy in normalizing the tumor microenvironment (TME) from an immunosuppressive to immunosupportive TME [38]. This may enhance the efficacy of all types of anticancer therapy, not only immunotherapy [39]. This is of particular relevance for *ALK* rearranged NSCLC, as the *EML4*-*ALK* gene fusion has been demonstrated previously to cause an immunosuppressive TME through activation of downstream signaling pathways such as PI3K, MAPK and Hippo pathways [40,41]. Upregulation of PD-L1 expression may occur through enhanced HIF1α expression as mentioned previously [31], with higher levels of PD-L1 expression in ALK positive versus negative tumors shown in several studies [40,42,43]. This points to a potential role for anti-angiogenic therapy in *ALK* rearranged NSCLC, whether in combination with immunotherapy or targeted therapy.

### 3.3. Other Pro-Angiogenic Factors or Alternative VEGF-Independent Angiogenic Pathways

The evidence to date for anti-angiogenic therapy in NSCLC has largely focused on the development of monoclonal antibodies and multi-kinase TKIs which target VEGF. However, emerging evidence suggests that compensatory mechanisms can occur with VEGF inhibition, leading to resistance. This may include increased levels of other pro-angiogenic factors or activation of alternative VEGF-independent angiogenic pathways [44]. Therefore, other targets or targeted therapies may also be considered in targeting angiogenesis pathways in NSCLC. To this end, multi-kinase TKIs may also act through complementary mechanisms, although the sensitivity for each target may differ, and similarly across different compounds. Therefore, it may be important to consider each multi-kinase TKI individually, in regard to kinase inhibition. Examples of other factors or alternative pathways include fibroblast growth factors (FGFs), which may induce angiogenesis through the enhanced proliferation and migration of endothelial cells [45]. Pre-clinical studies have demonstrated the increased expression of FGF and/or FGFR upon hypoxia in anti-VEGF resistant tumors [46,47]. Platelet-derived growth factor (PDGF) signaling leads to pro-angiogenic factor secretion, with subsequent increase in EC proliferation, migration, sprouting and tube formation [48]. Tumor revascularization due to PDGF signaling has been demonstrated in various tumors [49,50]. Hepatocyte growth factor (HGF)/c-MET, placental growth factor (PIGF) and angiopoietins are other pro-angiogenic factors which have demonstrated important roles in tumor angiogenesis [51,52,53]. Conversely, natural anti-angiogenic factors also represent potential therapeutic agents. These include thrombospondin, pigment epithelium-derived factor (PEDF) and endostatin, a C-terminal fragment of type XVVIII collagen [54]. In particular, a human recombinant endostatin, Endostar, has been evaluated in numerous clinical trials for NSCLC, demonstrating efficacy in combination with chemotherapy [55,56].

## 4. Clinical Evidence for Anti-Angiogenic Therapy in *ALK* Rearranged NSCLC

### 4.1. Anti-Angiogenic Therapy in Unselected NSCLC Populations

Several anti-angiogenic therapies have been evaluated in NSCLC. Most prominently, this includes monoclonal antibodies targeting VEGF such as bevacizumab and ramucirumab. Bevacizumab is a humanized monoclonal antibody which targets VEGF-A, whilst ramucirumab is a humanized monoclonal antibody which selectively targets VEGFR2 [57]. In addition, multi-kinase TKIs have been tested in NSCLC, such as nintedanib. Nintedanib targets several major angiogenic pathways, with activity against the PDGF, FGF and VEGF receptor families [58]. Previous trials and meta-analyses have demonstrated the efficacy for combination anti-VEGF therapy with chemotherapy in both the first- and second-line treatment settings with improved progression-free survival (PFS) and objective response rates (ORR) compared with chemotherapy alone [16,59,60,61,62,63]. This includes first-line carboplatin and paclitaxel with bevacizumab [64] and second-line docetaxel with ramucirumab [65] and docetaxel with nintedanib [66]. This led to the regulatory approval of these anti-angiogenic therapies with efficacy demonstrated in randomized phase III trials. A wide range of other monoclonal antibodies such as aflibercept [67], and TKIs such as axitinib [68], cediranib [69,70,71], linifanib [72], motesanib [73], sorafenib [74], sunitinib [75] and vandetanib [76,77,78,79], were also evaluated in randomized trials for NSCLC in both the first- and later-line settings. A human recombinant endostatin, Endostar, has also demonstrated efficacy in numerous randomized trials for advanced NSCLC as mentioned above, and has received regulatory approval in China [80].

Ultimately, however, concerns relating to costs, toxicities and lack of overall survival (OS) benefit limited the reimbursement and utilization of anti-angiogenic therapies in routine clinical practice. For example, a meta-analysis by Raphael et al. [59] of phase III trials demonstrated improvement in ORR and PFS, but no significant effect on OS. In another meta-analysis by Li et al. [81] of 15 randomized controlled phase II and III trials of anti-angiogenic TKIs in combination with chemotherapy versus chemotherapy alone, PFS and ORR were significantly increased. However, there was no difference in OS. Furthermore, grade ≥3 toxicities and treatment-related deaths were significantly higher with the combination of anti-angiogenic therapy with chemotherapy. Many of the trials evaluating anti-angiogenic agents, however, were conducted in unselected NSCLC populations, or where the *EGFR*/*ALK* status was unknown or not tested. With the introduction of immunotherapy into the treatment paradigm, more recent trials have further demonstrated the efficacy of anti-angiogenic therapy in combination with chemoimmunotherapy.

### 4.2. Combination Anti-Angiogenic Therapy with Chemoimmunotherapy

IMpower150 was a three-arm randomized controlled phase III trial for treatment-naïve patients with advanced non-squamous NSCLC [82]. The regimens in the three arms consisted of bevacizumab/carboplatin/paclitaxel (BCP), atezolizumab/carboplatin/paclitaxel (ACP) and atezolizumab/bevacizumab/carboplatin/paclitaxel (ABCP). In *EGFR*/*ALK* negative patients, median PFS (HR 0.62, 95%CI 0.52–0.74) and median OS (HR 0.80, 95%CI 0.67–0.95) were both prolonged with ABCP compared with BCP [83]. There was no statistically significant difference in OS comparing the ACP and BCP arms (HR 0.84, 95%CI 0.71–1.00). Importantly, however, this trial allowed patients with *EGFR*/*ALK* alterations after prior TKI treatment, reported separately as a subgroup analysis [84]. There were a total of 40 patients with *ALK* rearrangements treated on the trial, including *n* = 11 in the ABCP group, *n* = 9 in the ACP group and *n* = 20 in the BCP group (Table 1). PFS results for *ALK* rearranged patients were reported only together with *EGFR* mutated patients (*n* = 108 total), with PFS improvement for ABCP compared with BCP (9.7 versus 6.1 months, HR 0.59, 95%CI 0.37–0.94) [82]. OS results for the *ALK* rearranged subgroup, however, were subsequently reported—although the small numbers in each arm likely contributed to wide confidence intervals. In 31 patients, ABCP compared with BCP demonstrated median OS of not reached (NR) versus 6.9 months (HR 0.47, 95%CI 0.15–1.48). In 29 patients, ACP compared with BCP demonstrated a median OS of 13.6 versus 6.9 months (HR 0.59, 95%CI 0.20–1.72). Subsequently, there have also been several case reports documenting the potential use of the ABCP regimen in the post ALK TKI setting [85,86]. However, there remains no other prospective evidence to date of combination anti-angiogenic therapy with chemoimmunotherapy [87].

### 4.3. Combination Anti-Angiogenic Therapy with Targeted Therapy

Anti-angiogenic therapy in combination with targeted therapy has been most comprehensively evaluated in *EGFR* mutated NSCLC. Combinations of erlotinib with bevacizumab [88] and erlotinib with ramucirumab [89] have both demonstrated superior outcomes compared with erlotinib alone in randomized phase III trials. Consequently, this could be considered a standard treatment option in the first-line setting for advanced *EGFR* mutated NSCLC. In *ALK* rearranged NSCLC, the reported evidence to date of combination anti-angiogenic therapy with ALK TKI is more limited. The combination of crizotinib plus bevacizumab has been reported in a small single-arm prospective observational study, which included 12 treatment-naïve patients with *ALK* rearrangements [90]. The ORR was 58.3% and the disease control rate (DCR) was 100%. After median follow-up of 42.8 months, median PFS was 13.9 months and median duration of response (DOR) was 14.8 months. Fatigue (28.6%) and rash (21.4%) were the most common side effects reported, however three patients discontinued due to liver toxicity or hemoptysis. More recently, alectinib plus bevacizumab has also been evaluated in two single-arm trials. A phase I/II study of alectinib plus bevacizumab utilized a dose de-escalation strategy in the phase I portion of the study [91]. There were no dose-limiting toxicities at the standard doses of both drugs (alectinib 600 mg oral twice daily and bevacizumab 15 mg/kg intravenously every 3 weeks), and this was determined as the recommended phase II dose (RP2D). In total, there were 11 patients treated on the study, of which six (55%) were treatment-naïve, with the remaining patients having received prior ALK TKI but were alectinib-naïve. Median PFS was not reached and was 9.5 months in the treatment-naïve and ALK TKI pre-treated patients, respectively. Grade 3 or higher adverse events included pneumonitis (9%), proteinuria (9%), and hypertension (9%). The study, however, was closed prematurely due to slow accrual. Another single-arm phase II study of alectinib plus bevacizumab was conducted in Japan, using the standard local dose of alectinib (300 mg oral twice daily) with bevacizumab (15 mg/kg every 3 weeks) [92]. Patients in this study (*n* = 12) had progressed on prior alectinib, with 11 (92%) also having received prior crizotinib. The median PFS was 3.1 months with an ORR of 8% and DCR of 67%. Grade 3 or higher adverse events included anemia (8%), proteinuria (8%), diarrhea (8%) and hypokalemia (8%). A further single-arm phase II trial of alectinib and bevacizumab is currently ongoing (NCT03779191), however results are yet to be reported. Lastly, the combination of lorlatinib plus bevacizumab has been reported in a small case series of two patients resistant to prior ALK TKI, with disease regression and disease control (although with symptomatic improvement) after combination treatment, respectively [93]. The combination treatment was well tolerated, with bevacizumab given at doses of 15 mg/kg and 10 mg/kg every 3 weeks, respectively.

**Table 1 ijms-23-08863-t001:** Clinical trials evaluating anti-angiogenic therapy in *ALK* rearranged NSCLC.

Study	Study Type	Study Population (No. of Patients)	Treatment	ORR (%)	Median PFS (Months)	Median OS (Months)	Ref.
IMpower150	Phase III RCT	ALK TKI pre-treated (*n* = 40)	ABCP versus BCP versus ACP	-	-	ABCP versus BCP—NR versus 6.9 (HR 0.47, 95%CI 0.15–1.48)	[84]
						BCP versus ACP—13.6 versus 6.9 (HR 0.59, 95%CI 0.20–1.72)	
Lin et al., 2021	Phase I/II	Treatment-naïve (*n* = 6) and ALK TKI pre-treated but alectinib naïve (*n* = 5)	Alectinib 600 mg PO BD plus bevacizumab IV q3w	82	Treatment naïve cohort—NR ALK TKI pre-treated cohort—9.5	-	[91]
Watanabe et al., 2019	Single-arm phase II	Alectinib pre-treated (*n* = 12)	Alectinib 300 mg PO BD plus bevacizumab 15 mg/kg IV q3w	8	3.1	-	[92]
Huang et al., 2021	Single-arm observational	Treatment-naïve (*n* = 12)	Crizotinib 250 mg PO BD plus bevacizumab 7.5 mg/kg IV q3w	58.3	13.9	-	[90]

ABCP—atezolizumab, bevacizumab, carboplatin, paclitaxel; ACP—atezolizumab, carboplatin, paclitaxel; ALK—anaplastic lymphoma kinase; BCP—bevacizumab, carboplatin, paclitaxel; NR—not reached; ORR—objective response rate; OS—overall survival; PFS—progression-free survival; TKI—tyrosine kinase inhibitor.

## 5. Future Directions to Evaluate Anti-Angiogenic Therapy in *ALK* Rearranged NSCLC

Despite growing evidence for the efficacy of anti-angiogenic therapy in non-oncogene driven NSCLC and *EGFR* mutated NSCLC, there remain limited data on the efficacy of anti-angiogenic therapy in *ALK* rearranged patients. Prospective data are scarce, and difficulties in recruiting patients to trials for a relatively uncommon driver alteration is compounded by the large number of potential combination approaches with targeted therapies and chemoimmunotherapy. Interestingly, the ABCP regimen has received regulatory approval for *ALK* rearranged (and *EGFR* mutated) patients in Europe and Japan after prior TKI failure [83]. Although not statistically powered to detect differences between arms in the IMpower150 trial, the reported OS data in *ALK* rearranged patients suggest the added efficacy with bevacizumab may be limited. This emphasizes the need for larger datasets and greater evidence to evaluate the four-drug regimen, where costs and toxicities remain a prominent consideration. Similarly, the prospective evidence for combination ALK TKI with bevacizumab suggests only modest efficacy, particularly in ALK TKI pre-treated patients. Several ongoing trials will add to the evidence base. This includes the aforementioned single-arm trial of alectinib plus bevacizumab (NCT03779191), as well as a phase I trial of brigatinib plus bevacizumab (NCT04227028). In addition, a phase I trial evaluating the combination of ensartinib with carboplatin, pemetrexed and bevacizumab (NCT04837716) will provide preliminary evidence on the novel combination of ALK TKI with anti-angiogenic therapy and chemotherapy. This trial will use a dose de-escalation strategy, followed by a dose-expansion study. A note of caution, however, may be observed from trials evaluating the combination of ALK TKI with immunotherapy, in which toxicity was significantly increased, without signs of greater efficacy [94,95]. Whilst results from these ongoing trials will provide additional data, sample sizes are likely to be small. This highlights the need for alternative and innovative methods of generating high-quality and robust clinical evidence. Real-world evidence and big data initiatives may play a broader role in complementing clinical trial data in this regard [96].

Nevertheless, despite modest evidence to date, in selected patients anti-angiogenic therapy may still have an important role. Given the propensity for *ALK* rearranged NSCLC to develop central nervous system (CNS) metastases, and the activity of bevacizumab for radiation necrosis [97], this may be a specific subset of patients that may benefit from bevacizumab therapy. A small case series has demonstrated the potential efficacy of bevacizumab in combination with continued ALK TKI in managing brain radiation necrosis [98].

A further consideration apart from the optimal combination is the optimal line of therapy and therapeutic sequencing [99]. With several generations of ALK TKI, the treatment landscape is increasingly more complex. After progression on first-line ALK TKI, treatment options may include sequencing with next-generation ALK TKI, chemotherapy or chemoimmunotherapy, or even rechallenge with prior ALK TKI. Potentially, anti-angiogenic therapy could be considered in combination with all of these options, whether as an on- or off-label treatment. Emerging data relating to on-target and off-target mechanisms of resistance further add to the complexity for treatment decision making [100]. Relatedly, a case report suggested bevaciuzumab may have reversed primary resistance to alectinib in a patient after combined bevacizumab plus chemotherapy treatment [101]. The patient subsequently responded to rechallenge with alectinib. Interestingly, in this case, patient-derived cell lines taken before and after initial alectinib therapy were sensitive to alectinib ex vivo. Consequently, it was hypothesized that bevacizumab may have normalized the tumor vasculature to allow improved delivery of alectinib into the tumor, suggesting that the TME may have been responsible for the initial primary resistance to alectinib. In this context, translational research for predictive biomarkers for anti-angiogenic therapy efficacy becomes of paramount importance. Rigorous pre-clinical and translational studies are needed to better elucidate key patient and tumor characteristics that may predict for response and identify rational combination approaches.

Ultimately, with prospective clinical trial data likely to be limited, greater pre-clinical and translational evidence will be crucial to elucidate the role for anti-angiogenic therapy in *ALK* rearranged NSCLC. We have increasing understanding of the molecular heterogeneity of lung cancer. In particular, stark underlying biologic differences of oncogenic driven lung cancers with gene rearrangements compared to activating gene mutations [102] point towards a need to consider these subgroups separately. Finally, high-quality real-world studies may also represent the most feasible method to generate clinical evidence in this area of unmet need.

## 6. Conclusions

There is now a large body of evidence supporting the role of anti-angiogenic therapy in the treatment paradigm for advanced lung cancer. For *ALK* rearranged NSCLC however, the evidence is limited, and prospective clinical data suggest only modest efficacy. Greater evidence to support the use of anti-angiogenic therapy in *ALK* rearranged NSCLC is needed, including pre-clinical, translational, and clinical data.

## Figures and Tables

**Figure 1 ijms-23-08863-f001:**
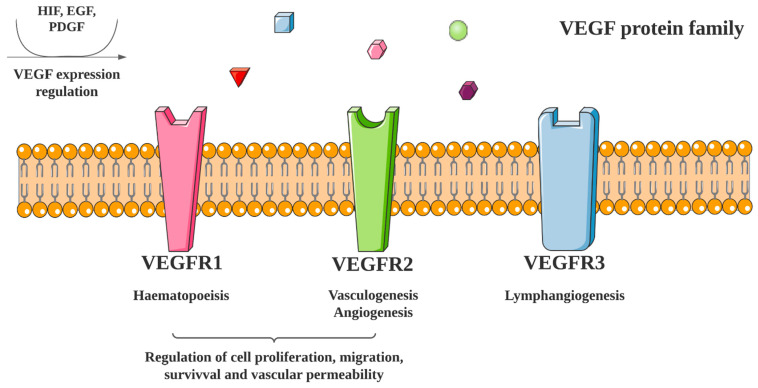
VEGF activation and signaling pathway. The VEGF protein family includes VEGF-A, -B, -C, -D, and -E. VEGF-A binds to VEGFR1 homo-, VEGFR2 homo- and VEGFR1/R2 hetero-dimers, VEGF-B binds to VEGFR1 homodimers, VEGF-C and -D bind to VEGFR3 homodimers and VEGF-E binds to VEGFR2 homodimers.

**Figure 2 ijms-23-08863-f002:**
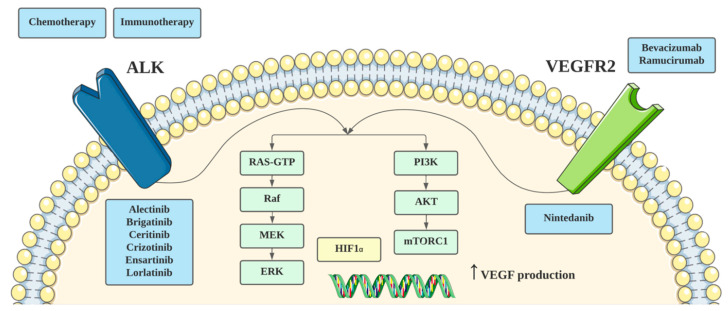
Inhibition of and potential cross-talk between the VEGF and ALK pathways.

## Data Availability

Not applicable.

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
