# Peer review of "Anti-Angiogenic Therapy in ALK Rearranged Non-Small Cell Lung Cancer (NSCLC)"

_ijms, 2022, doi:10.3390/ijms23168863_

Round 1

Reviewer 1 Report

The paper is a brief but comprehensive review on the potential of anti-angioigenic therapies for the treatment non-small cell lung cancer (NSCLC) with anaplastic lymphoma kinase 11 (ALK) gene rearrangements.

I have no major comments about this paper. However, I think it would benefit if the authors, instead of going straight to the description of the existing therapies, started the Introduction with a brief description of the ALK rearrangements (e.g., most common fusion partners) and their effects on the kinase activity, as well as clinical significance.

Also the Figure 1 would benefit from a more descriptive figure legend, more precisely: concerning the affinity of individual VEGFs for different receptor subtypes.

Authors should also make minor corrections. For example, in lines 51-52 a more appropriate word would be “heterodimers” instead of “heterodimerisation”; the ABCG2, OS, ORR and PFS abbreviations should be explained when they appear for the first time in the text.

Line 255: “Despite growing evidence for anti-angiogenic therapy in non-oncogene driven NSCLC...” – the word “efficacy” is missing (evidence for efficacy).

After these minor changes, the paper can be accepted for publication.

Reviewer 2 Report

Tan and Pavlakis reviewed the current status and future opportunities for anti-angiogenic therapy in ALK-rearranged NSCLC. They briefly highlighted VEGF and other angiogenic factors and their role in angiogenesis, preclinical studies of anti-angiogenic therapy, and clinical studies including combinatorial approaches with targeted and immunotherapeutics. Overall, the review is well-written and is covers an interesting area of research in NSCLC therapy; thus, I recommend the review for publication in the journal
